# Text Embeddings Reveal (Almost) As Much As Text

**John X. Morris, Volodymyr Kuleshov, Vitaly Shmatikov, Alexander M. Rush**
Department of Computer Science
Cornell University

## Abstract

How much private information do text embeddings reveal about the original text? We investigate the problem of embedding *inversion*, reconstructing the full text represented in dense text embeddings. We frame the problem as controlled generation: generating text that, when reembedded, is close to a fixed point in latent space. We find that although a naïve model conditioned on the embedding performs poorly, a multi-step method that iteratively corrects and re-embeds text is able to recover 92% of 32-token text inputs exactly. We train our model to decode text embeddings from two state-of-the-art embedding models, and also show that our model can recover important personal information (full names) from a dataset of clinical notes. [1]

## 1 Introduction

Systems that utilize large language models (LLMs) often store auxiliary data in a vector database of dense embeddings (Borgeaud et al., 2022; Yao et al., 2023). Users of these systems infuse knowledge into LLMs by inserting retrieved documents into the language model's prompt. Practitioners are turning to hosted vector database services to execute embedding search efficiently at scale (Pinecone; Qdrant; Vdaas; Weaviate; LangChain). In these databases, the data owner only sends *embeddings* of text data (Le and Mikolov, 2014; Kiros et al., 2015) to the third party service, and never the text itself. The database server returns a search result as the index of the matching document on the client side.

Vector databases are increasingly popular, but privacy threats within them have not been comprehensively explored. Can the third party service to reproduce the initial text, given its embedding? Neural networks are in general non-trivial or even

impossible to invert exactly. Furthermore, when querying a neural network through the internet, we may not have access to the model weights or gradients at all.

Still, given input-output pairs from a network, it is often possible to approximate the network's inverse. Work on *inversion* in computer vision (Mahendran and Vedaldi, 2014; Dosovitskiy and Brox, 2016) has shown that it is possible to learn to recover the input image (with some loss) given the logits of the final layer. Preliminary work has explored this question for text (Song and Raghunathan, 2020), but only been able to recover an approximate bag of words given embeddings from shallow networks.

In this work, we target full reconstruction of input text from its embedding. If text is recoverable, there is a threat to privacy: a malicious user with access to a vector database, and text-embedding pairs from the model used to produce the data, could learn a function that reproduces text from embeddings.

We frame this problem of recovering textual embeddings as a controlled generation problem, where we seek to generate text such that the text is as close as possible to a given embedding. Our method, *Vec2Text*, uses the difference between a hypothesis embedding and a ground-truth embedding to make discrete updates to the text hypothesis.

When we embed web documents using a state-of-the-art black-box encoder, our method can recover 32-token inputs with a near-perfect BLEU score of 97.3, and can recover 92% of the examples exactly. We then evaluate on embeddings generated from a variety of common retrieval corpuses from the BEIR benchmark. Even though these texts were not seen during training, our method is able to perfectly recover the inputs for a number of datapoints across a variety of domains. We evaluate on embeddings of clinical notes from MIMIC and are able to recover 89% of full names from embedded

---

[1] Our code is available on Github: github.com/jxmorris12/vec2text.

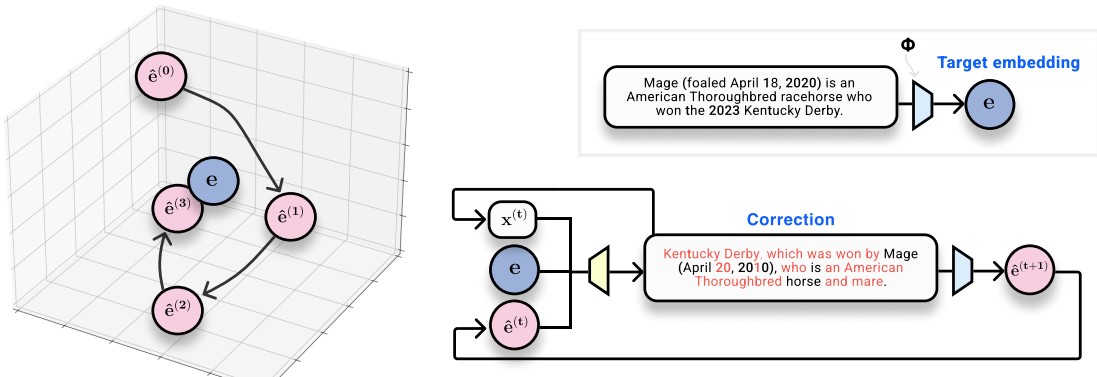

Figure 1: Overview of our method, Vec2Text. Given access to a target embedding $e$ (blue) and query access to an embedding model $\phi$ (blue model), the system aims to iteratively generate (yellow model) hypotheses $\hat{e}$ (pink) to reach the target. Example input is a taken from a recent Wikipedia article (June 2023). Vec2Text perfectly recovers this text from its embedding after 4 rounds of correction.

notes. These results imply that text embeddings present the same threats to privacy as the text from which they are computed, and embeddings should be treated with the same precautions as raw data.

## 2   Overview: Embedding Inversion

Text embedding models learn to map text sequences to embedding vectors. Embedding vectors are useful because they encode some notion of semantic similarity: inputs that are similar in meaning should have embeddings that are close in vector space (Mikolov et al., 2013). Embeddings are commonly used for many tasks such as search, clustering, and classification (Aggarwal and Zhai, 2012; Neelakantan et al., 2022; Muennighoff et al., 2023).

Given a text sequence of tokens $x \in \mathbb{V}^n$, a text encoder $\phi : \mathbb{V}^n \to \mathbb{R}^d$ maps $x$ to a fixed-length embedding vector $e \in \mathbb{R}^d$.

Now consider the problem of inverting textual embeddings: given some unknown encoder $\phi$, we seek to recover the text $x$ given its embedding $e = \phi(x)$. Text embedding models are typically trained to encourage similarity between related inputs (Karpukhin et al., 2020). Thus, we can write the problem as recovering text that has a maximally similar embedding to the ground-truth. We can formalize the search for text $\hat{x}$ with embedding $e$ under encoder $\phi$ as optimization:

$$\hat{x} = \arg\max_x \cos(\phi(x), e) \qquad (1)$$

**Assumptions of our threat model.**   In a practical sense, we consider the scenario where an attacker wants to invert a single embedding produced from a black-box embedder $\phi$. We assume that the attacker has access to $\phi$: given hypothesis text $\hat{x}$, the attacker can query the model for $\phi(\hat{x})$ and compute $\cos(\phi(\hat{x}), e)$. When this term is 1 exactly, the attacker can be sure that $\hat{x}$ was the original text, i.e. collisions are rare and can be ignored.

## 3   Method: Vec2Text

### 3.1   Base Model: Learning to Invert $\phi$

Enumerating all possible sequences to compute Equation (1) is computationally infeasible. One way to avoid this computational constraint is by learning a distribution of texts given embeddings. Given a dataset of texts $\mathcal{D} = \{x_1, \ldots\}$, we learn to invert encoder $\phi$ by learning a distribution of texts given embeddings, $p(x \mid e; \theta)$, by learning $\theta$ via maximum likelihood:

$$\theta = \arg\max_{\hat{\theta}} \mathbb{E}_{x \sim \mathcal{D}}[p(x \mid \phi(x); \hat{\theta})]$$

We drop the $\theta$ hereon for simplicity of notation. In practice, this process involves training a conditional language model to reconstruct unknown text $x$ given its embedding $e = \phi(x)$. We can view this learning problem as amortizing the combinatorial optimization (Equation (1)) into the weights of a neural network. Directly learning to generate satisfactory text in this manner is well-known in the literature to be a difficult problem.

## 3.2 Controlling Generation for Inversion

To improve upon this model, we propose Vec2Text shown in Figure 1. This approach takes inspiration from methods for Controlled Generation, the task of generating text that satisfies a known condition (Hu et al., 2018; John et al., 2018; Yang and Klein, 2021). This task is similar to inversion in that there is a observable function $\phi$ that determines the level of control. However, it differs in that approaches to controlled generation (Dathathri et al., 2020; Li et al., 2022) generally require differentiating through $\phi$ to improve the score of some intermediate representation. Textual inversion differs in that we can only make queries to $\phi$, and cannot compute its gradients.

**Model.** The method guesses an initial hypothesis and iteratively refines this hypothesis by re-embedding and correcting the hypothesis to bring its embedding closer to $e$. Note that this model requires computing a new embedding $\hat{e}^{(t)} = \phi(x^{(t)})$ in order to generate each new correction $x^{(t+1)}$. We define our model recursively by marginalizing over intermediate hypotheses:

$$p(x^{(t+1)} \mid e) = \sum_{x^{(t)}} p(x^{(t)} \mid e) p(x^{(t+1)} \mid e, x^{(t)}, \hat{e}^{(t)})$$
$$\hat{e}^{(t)} = \phi(x^{(t)})$$

with a base case of the simple learned inversion:

$$p(x^{(0)} \mid e) = p(x^{(0)} \mid e, \varnothing, \phi(\varnothing))$$

Here, $x^{(0)}$ represents the initial hypothesis generation, $x^{(1)}$ the correction of $x^{(0)}$, and so on. We train this model by first generating hypotheses $x^{(0)}$ from the model in Section 3.1, computing $\hat{e}^{(0)}$, and then training a model on this generated data.

This method relates to other recent work generating text through iterative editing (Lee et al., 2018; Ghazvininejad et al., 2019). Especially relevant is Welleck et al. (2022), which proposes to train a text-to-text 'self-correction' module to improve language model generations with feedback.

**Parameterization.** The backbone of our model, $p(x^{(t+1)} \mid e, x^{(t)}, \hat{e}^{(t)})$, is parameterized as a standard encoder-decoder transformer (Vaswani et al., 2017; Raffel et al., 2020) conditioned on the previous output.

One challenge is the need to input conditioning embeddings $e$ and $\hat{e}^{(t)}$ into a transformer encoder, which requires a sequence of embeddings as input with some dimension $d_{\text{enc}}$ not necessarily equal to the dimension $d$ of $\phi$'s embeddings. Similar to Mokady et al. (2021), we use small MLP to project a single embedding vector to a larger size, and reshape to give it a sequence length as input to the encoder. For embedding $e \in \mathbb{R}^d$:

$$\text{EmbToSeq}(e) = W_2 \, \sigma(W_1 \, e)$$

where $W_1 \in \mathbb{R}^{d \times d}$ and $W_2 \in \mathbb{R}^{(sd_{\text{enc}}) \times d}$ for some nonlinear activation function $\sigma$ and predetermined encoder "length" $s$. We use a separate MLP to project three vectors: the ground-truth embedding $e$, the hypothesis embedding $\hat{e}^{(t)}$, and the difference between these vectors $e - \hat{e}$. Given the word embeddings of the hypothesis $x^{(t)}$ are $\{w_1...w_n\}$, the input (length $3s + n$) to the encoder is as follows:

$$\begin{aligned}\text{concat}(&\text{EmbToSeq}(e), \\ &\text{EmbToSeq}(\hat{e}^{(t)}), \\ &\text{EmbToSeq}(e - \hat{e}^{(t)}), (w_1...w_n))\end{aligned}$$

We feed the concatenated input to the encoder and train the full encoder-decoder model using standard language modeling loss.

**Inference.** In practice we cannot tractably sum out intermediate generations $x^{(t)}$, so we approximate this summation via beam search. We perform inference from our model greedily at the token level but implement beam search at the sequence-level $x^{(t)}$. At each step of correction, we consider some number $b$ of possible corrections as the next step. For each possible correction, we decode the top $b$ possible continuations, and then take the top $b$ unique continuations out of $b \cdot b$ potential continuations by measuring their distance in embedding space to the ground-truth embedding $e$.

## 4 Experimental Setup

**Embeddings.** Vec2Text is trained to invert two state-of-the-art embedding models: GTR-base (Ni et al., 2021), a T5-based pre-trained transformer for text retrieval, and text-embeddings-ada-002 available via the OpenAI API. Both model families are among the highest-performing embedders on the MTEB text embeddings benchmark (Muennighoff et al., 2023).

| | method | tokens | pred tokens | bleu | tf1 | exact | cos |
|---|---|---|---|---|---|---|---|
| **GTR Natural Questions** | Bag-of-words (Song and Raghunathan, 2020) | 32 | 32 | 0.3 | 51 | 0.0 | 0.70 |
| | GPT-2 Decoder (Li et al., 2023) | 32 | 32 | 1.0 | 47 | 0.0 | 0.76 |
| | Base [0 steps] | 32 | 32 | 31.9 | 67 | 0.0 | 0.91 |
| | (+ beam search) | 32 | 32 | 34.5 | 67 | 1.0 | 0.92 |
| | (+ nucleus) | 32 | 32 | 25.3 | 60 | 0.0 | 0.88 |
| | Vec2Text [1 step] | 32 | 32 | 50.7 | 80 | 0.0 | 0.96 |
| | [20 steps] | 32 | 32 | 83.9 | 96 | 40.2 | 0.99 |
| | [50 steps] | 32 | 32 | 85.4 | 97 | 40.6 | 0.99 |
| | [50 steps + sbeam] | 32 | 32 | **97.3** | 99 | 92.0 | **0.99** |
| **OpenAI MSMARCO** | Base [0 steps] | 31.8 | 31.8 | 26.2 | 61 | 0.0 | 0.94 |
| | Vec2Text [1 step] | 31.8 | 31.9 | 44.1 | 77 | 5.2 | 0.96 |
| | [20 steps] | 31.8 | 31.9 | 61.9 | 87 | 15.0 | 0.98 |
| | [50 steps] | 31.8 | 31.9 | 62.3 | 87 | 14.8 | 0.98 |
| | [50 steps + sbeam] | 31.8 | 31.8 | **83.4** | 96 | 60.9 | **0.99** |
| **OpenAI MSMARCO** | Base [0 steps] | 80.9 | 84.2 | 17.0 | 54 | 0.6 | 0.95 |
| | Vec2Text [1 step] | 80.9 | 81.6 | 29.9 | 68 | 1.4 | 0.97 |
| | [20 steps] | 80.9 | 79.7 | 43.1 | 78 | 3.2 | 0.99 |
| | [50 steps] | 80.9 | 80.5 | 44.4 | 78 | 3.4 | 0.99 |
| | [50 steps + sbeam] | 80.9 | 80.6 | **55.0** | 84 | 8.0 | **0.99** |

Table 1: Reconstruction score on in-domain datasets. Top section of results come from models trained to reconstruct 32 tokens of text from Wikpedia, embedded using GTR-base. Remaining results come from models trained to reconstruct up to 32 or 128 tokens from MSMARCO, embedded using OpenAI `text-embeddings-ada-002`.

**Datasets.** We train our GTR model on $5M$ passages from Wikipedia articles selected from the Natural Questions corpus (Kwiatkowski et al., 2019) truncated to 32 tokens. We train our two OpenAI models (Bajaj et al., 2018), both on versions of the MSMARCO corpus with maximum 32 or 128 tokens per example [2]. For evaluation, we consider the evaluation datasets from Natural Questions and MSMarco, as well as two out-of-domain settings: the MIMIC-III database of clinical notes (Johnson et al., 2016) in addition to the variety of datasets available from the BEIR benchmark (Thakur et al., 2021).

**Baseline.** As a baseline, we train the base model $p(x^{(0)} \mid e)$ to recover text with no correction steps. We also evaluate the bag of words model from Song and Raghunathan (2020). To balance for the increased number of queries allotted to the correction models, we also consider taking the top-N predictions made from the unconditional model via beam search and nucleus sampling ($p = 0.9$) and reranking via cosine similarity.

**Metrics.** We use two types of metrics to measure the progress and the accuracy of reconstructed text. First we consider our main goal of text reconstruction. To measure this we use word-match metrics including: BLEU score (Papineni et al., 2002), a measure of n-gram similarities between the true and reconstructed text; Token F1, the multi-class F1 score between the set of predicted tokens and the set of true tokens; Exact-match, the percentage of reconstructed outputs that perfectly match the ground-truth. We also report the similarity on the internal inversion metric in terms of recovering the vector embedding in latent space. We use cosine similarity between the true embedding and the embedding of reconstructed text according to $\phi$.

**Models and Inference.** We initialize our models from the T5-base checkpoint (Raffel et al., 2020). Including the projection head, each model has approximately 235M parameters. We set the projection sequence length $s = 16$ for all experiments, as preliminary experiments show diminishing returns by increasing this number further. We perform inference on all models using greedy token-level decoding. When running multiple steps of

[2]By 2023 pricing of $0.0001 per 1000 tokens, embedding 5 million documents of 70 tokens each costs $35.

| dataset | tokens | method | bleu | token F1 |
|---|---|---|---|---|
| quora | 15.7 | Base | 36.2 | 73.8 |
| | | Vec2Text | 95.5 | 98.6 |
| signal1m | 23.7 | Base | 13.2 | 49.5 |
| | | Vec2Text | 80.7 | 92.5 |
| msmarco | 72.1 | Base | 15.5 | 54.1 |
| | | Vec2Text | 59.6 | 86.1 |
| climate-fever | 73.4 | Base | 12.8 | 49.3 |
| | | Vec2Text | 44.9 | 82.6 |
| fever | 73.4 | Base | 12.6 | 49.2 |
| | | Vec2Text | 45.1 | 82.7 |
| dbpedia-entity | 91.3 | Base | 15.4 | 50.3 |
| | | Vec2Text | 48.0 | 77.9 |
| nq | 94.7 | Base | 11.0 | 47.1 |
| | | Vec2Text | 32.7 | 72.7 |
| hotpotqa | 94.8 | Base | 15.4 | 50.1 |
| | | Vec2Text | 46.6 | 78.7 |
| fiqa | 103.8 | Base | 6.6 | 44.1 |
| | | Vec2Text | 21.5 | 63.6 |
| webis-touche2020 | 105.2 | Base | 6.6 | 41.5 |
| | | Vec2Text | 19.6 | 69.7 |
| cqadupstack | 106.4 | Base | 7.1 | 41.5 |
| | | Vec2Text | 23.3 | 64.3 |
| arguana | 113.5 | Base | 6.8 | 44.1 |
| | | Vec2Text | 23.4 | 66.3 |
| scidocs | 125.3 | Base | 5.9 | 38.5 |
| | | Vec2Text | 17.7 | 57.6 |
| trec-covid | 125.4 | Base | 5.6 | 36.3 |
| | | Vec2Text | 19.3 | 58.6 |
| robust04 | 127.3 | Base | 4.9 | 34.4 |
| | | Vec2Text | 15.5 | 54.5 |
| bioasq | 127.4 | Base | 5.3 | 35.7 |
| | | Vec2Text | 22.8 | 59.5 |
| scifact | 127.4 | Base | 4.9 | 35.2 |
| | | Vec2Text | 16.6 | 56.6 |
| nfcorpus | 127.7 | Base | 6.2 | 39.6 |
| | | Vec2Text | 25.8 | 64.8 |
| trec-news | 128.0 | Base | 4.9 | 34.8 |
| | | Vec2Text | 14.5 | 51.5 |

Table 2: Out-of-domain reconstruction performance measured on datasets from the BEIR benchmark. We sort datasets in order of average length in order to emphasize the effect of sequence length on task difficulty.

sequence-level beam search, we only take a new generation if it is closer than the previous step in cosine similarity to the ground-truth embedding.

We use unconditional models to seed the initial hypothesis for our iterative models. We examine the effect of using a different initial hypothesis in Section 7.

We use the Adam optimizer and learning rate of

$2 * 10^{-4}$ with warmup and linear decay. We train models for 100 epochs. We use batch size of 128 and train all models on 4 NVIDIA A6000 GPUs. Under these conditions, training our slowest model takes about two days.

## 5 Results

### 5.1 Reconstruction: In-Domain

Table 1 contains in-domain results. Our method outperforms the baselines on all metrics. More rounds is monotonically helpful, although we see diminishing returns – we are able to recover 77% of BLEU score in just 5 rounds of correction, although running for 50 rounds indeed achieves a higher reconstruction performance. We find that running sequence-level beam search (sbeam) over the iterative reconstruction is particularly helpful for finding exact matches of reconstructions, increasing the exact match score by 2 to 6 times across the three settings. In a relative sense, the model has more trouble exactly recovering longer texts, but still is able to get many of the words.

### 5.2 Reconstruction: Out-of-Domain

We evaluate our model on 15 datasets from the BEIR benchmark and display results in Table 2. Quora, the shortest dataset in BEIR, is the easiest to reconstruct, and our model is able to exactly recover 66% of examples. Our model adapts well to different-length inputs, generally producing reconstructions with average length error of fewer than 3 tokens. In general, reconstruction accuracy inversely correlates with example length (discussed more in Section 7). On all datasets, we are able to recover sequences with Token F1 of at least 41 and cosine similarity to the true embedding of at least 0.95.

### 5.3 Case study: MIMIC

As a specific threat domain, we consider MIMIC-III clinical notes (Johnson et al., 2016). Because the original release of MIMIC is completely deidentified, we instead use the "pseudo re-identified" version from Lehman et al. (2021) where fake names have been inserted in the place of the deidentified ones.

Each note is truncated to 32 tokens and the notes are filtered so that they each contain at least one name. We measure the typical statistics of our method as well as three new ones: the percentage of first names, last names, and complete names

| method | first | last | full | bleu | tf1 | exact | cos |
|--------|-------|------|------|------|-----|-------|-----|
| Base | 40.0 | 27.8 | 10.8 | 4.9 | 33.1 | 0. | 0.78 |
| Vec2Text | 94.2 | 95.3 | 89.2 | 55.6 | 80.8 | 26.0 | 0.98 |

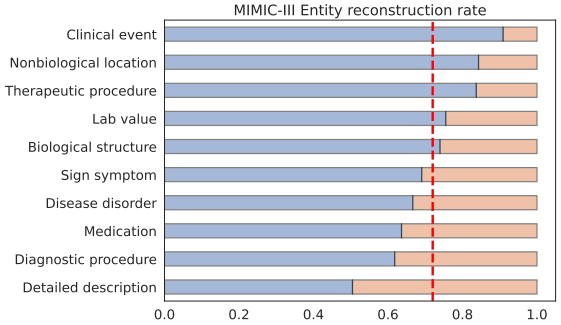

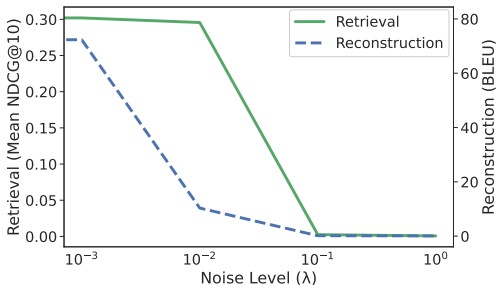

Figure 2: Retrieval performance and reconstruction accuracy across varying levels of noise injection.

Table 3: Performance of our method on reconstructing GTR-embedded clinical notes from MIMIC III (Johnson et al., 2016).

that are recovered. Results are shown in Table 3. Vec2Text is able to recover 94% of first names, 95% of last names, and 89% of full names (first, last format) while recovering 26% of the documents exactly.

For the recovered clinical notes from Section 5.3, we extract entities from each true and recovered note using a clinical entity extractor (Raza et al., 2022). We plot the recovery percentage in 3 (bottom) with the average entity recovery shown as a dashed line. Our model is most accurate at reconstructing entities of the type "Clinical Event", which include generic medical words like 'arrived', 'progress', and 'transferred'. Our model is least accurate in the "Detailed Description" category, which includes specific medical terminology like 'posterior' and 'hypoxic', as well as multi-word events like 'invasive ventilation - stop 4:00 pm'.

Although we are able to recover 26% of 32-token notes exactly, the notes that were not exactly recovered are semantically close to the original. Our model generally matches the syntax of notes, even when some entities are slightly garbled; for example, given the following sentence from a doctor's note "Rhona Arntson npn/- # resp: infant remains orally intubated on imv / r fi" our model predicts "Rhona Arpson nrft:# infant remains intubated orally on resp. imv. m/n fi".

## 6 Defending against inversion attacks

Is it easy for users of text embedding models protect their embeddings from inversion attacks? We consider a basic defense scenario as a sanity check. To implement our defense, the user addes a level of Gaussian noise directly to each embedding with the goal of effectively defending against inversion attacks while preserving utility in the nearest-neightbor retrieval setting. We analyze the trade-off between retrieval performance and reconstruction accuracy under varying levels of noise.

Formally, we define a new embedding model as:

$$\phi_{\text{noisy}}(x) = \phi(x) + \lambda \cdot \epsilon, \epsilon \sim N(0, 1)$$

where $\lambda$ is a hyperparameter controlling the amount of noise injected.

We simulate this scenario with $\phi$ as GTR-base using our self-corrective model with 10 steps of correction, given the noisy embedder $\phi_{\text{noisy}}$. To measure retrieval performance, we take the mean NDCG@10 (a metric of retrieval performance; higher is better) across 15 different retrieval tasks from the BEIR benchmark, evaluated across varying levels of noise.

We graph the average retrieval performance in Figure 2 (see A.2 for complete tables of results). At a noise level of $\lambda = 10^{-1}$, we see retrieval performance is preserved, while BLEU score drops by 10%. At a noise level of 0.01, retrieval performance is barely degraded (2%) while reconstruction performance plummets to 13% of the original BLEU. Adding any additional noise severely impacts both retrieval performance and reconstruction accuracy. These results indicate that adding a small amount of Gaussian noise may be a straightforward way to defend against naive inversion attacks, although it is possible that training with noise could in theory help Vec2Text recover more accurately from $\phi_{noisy}$. Note that low reconstruction BLEU score is not necessarily indicative that coarser inferences, such as clinical area or treatment regimen, cannot be made from embeddings.

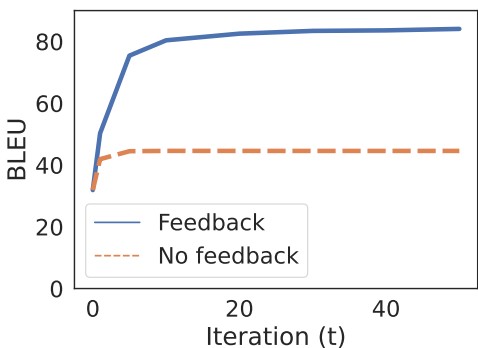

Figure 3: Recovery performance across multiple rounds of self-correction comparing models with access to $\phi$ vs text-only (32 tokens per sequence).

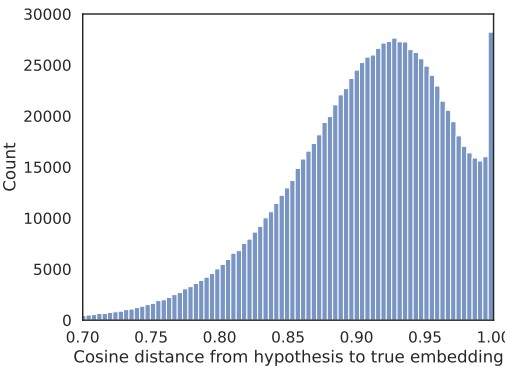

Figure 4: Distribution of $\cos(e, \phi(x^{(0)}))$ over training data. The mean training output from the GTR base model has a cosine similarity of $0.924$ with the true embedding.

## 7 Analysis

**How much does the model rely on feedback from $\phi$?** Figure 3 shows an ablation study of the importance of feedback, i.e. performing corrections with and without embedding the most recent hypothesis. The model trained with feedback (i.e. additional conditioning on $\phi(x^{(t)})$ is able to make a more accurate first correction and gets better BLEU score with more rounds. The model trained with no feedback can still edit the text but does not receive more information about the geometry of the embedding space, and quickly plateaus. The most startling comparison is in terms of the number of exact matches: after 50 rounds of greedy self-correction, our model with feedback gets $52.0\%$ of examples correct (after only $1.5\%$ initially); the model trained without feedback only perfectly matches $4.2\%$ of examples after 50 rounds.

During training, the model only learns to correct a single hypothesis to the ground-truth sample. Given new text at test time, our model is able to correct the same text multiple times, "pushing" the text from 0.9 embedding similarity to 1.0. We plot the closeness of the first hypothesis to the ground-truth in the training data for the length-32 model in Figure 4. We see that during training the model learns to correct hypotheses across a wide range of closenesses, implying that corrections should not go 'out-of-distribution' as they approach the ground-truth.

**How informative are embeddings for textual recovery?** We graph BLEU score vs. cosine similarity from a selection of of reconstructed text inputs in Figure 5. We observe a strong correlation between the two metrics. Notably, there are very

few generated samples with high cosine similarity but low BLEU score. This implies that better following embedding geometry will further improves systems. Theoretically some embeddings might be impossible to recover. Prior work (Song et al., 2020; Morris, 2020) has shown that two different sequences can 'collide' in text embedding space, having similar embeddings even without any word overlap. However, our experiments found no evidence that collisions are a problem; they either do not exist or our model learns during training to avoid outputting them. Improved systems should be able to recover longer text.

**Does having a strong base model matter?** We ablate the impact of initialization by evaluating our 32-token Wikipedia model at different initializations of $x^{(0)}$, as shown in Section 7. After running for 20 steps of correction, our model is able to recover from an unhelpful initialization, even when the initialization is a random sequence of tokens. This suggests that the model is able to ignore bad hypotheses and focus on the true embedding when the hypothesis is not helpful.

## 8 Related work

**Inverting deep embeddings.** The task of inverting textual embeddings is closely related to research on inverting deep visual representations in computer vision (Mahendran and Vedaldi, 2014; Dosovitskiy and Brox, 2016; Teterwak et al., 2021; Bordes et al., 2021), which show that a high amount of visual detail remains in the logit vector of an image classifier, and attempt to reconstruct input images from this vector. There is also a line of work reverse-engineering the content of certain text em-

| Input | Nabo Gass (25 August, 1954 in Ebingen, Germany) is a German painter and glass artist. | |
|---|---|---|
| Round 1 (0.85): | Nabo Gass (11 August 1974 in Erlangen, Germany) is an artist. | ✗ |
| Round 2 (0.99): | Nabo Gass (b. 18 August 1954 in Egeland, Germany) is a German painter and glass artist. | ✗ |
| Round 3 (0.99): | Nabo Gass (25 August 1954 in Ebingen, Germany) is a German painter and glass artist. | ✗ |
| Round 4 (1.00): | Nabo Gass (25 August, 1954 in Ebingen, Germany) is a German painter and glass artist. | ✓ |

Table 4: Example of our corrective model working in multiple rounds. Left column shows the correction number, from Round 1 (initial hypothesis) to Round 4 (correct guess). The number in parenthesis is the cosine similarity between the guess's embedding and the embedding of the ground-truth sequence (first row).

| Initialization | token f1 | cos | exact |
|---|---|---|---|
| Random tokens | 0.95 | 0.99 | 50.0 |
| "the " * 32 | 0.95 | 0.99 | 49.8 |
| "there's no reverse on a motorcycle, as my friend found out quite dramatically the other day" | 0.96 | 0.99 | 52.0 |
| Base model $p(x^{(0)} \mid e)$ | 0.96 | 0.99 | 51.6 |

Table 5: Ablation: Reconstruction score on Wikipedia data (32 tokens) given various initializations. Our self-correction model is able to faithfully recover the original text with greater than 80 BLEU score, even with a poor initialization. Models run for 20 steps of correction.

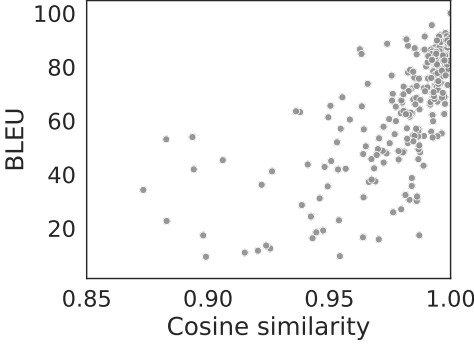

Figure 5: Cosine similarity vs BLEU score on 1000 reconstructed embeddings from Natural Questions text.

beddings: Ram et al. (2023) analyze the contents of text embeddings by projecting embeddings into the model's vocabulary space to produce a distribution of relevant tokens. Adolphs et al. (2022) train a single-step query decoder to predict the text of queries from their embeddings and use the decoder to produce more data to train a new retrieval model. We focus directly on text reconstruction and its implications for privacy, and propose an iterative method that works for paragraph-length documents, not just sentence-length queries.

**Privacy leakage from embeddings.** Research has raised the question of information leakage from dense embeddings. In vision, Vec2Face (Duong et al., 2020) shows that faces can be reconstructed from their deep embeddings. Similar questions have been asked about text data: Lehman et al. (2021) attempt to recover sensitive information such as names from representations obtained from a model pre-trained on clinical notes, but fail to recover exact text. Kim et al. (2022) propose a privacy-preserving similarity mechanism for text embeddings and consider a shallow bag-of-words inversion model. Abdalla et al. (2020) analyze the privacy leaks in training word embeddings on medical data and are able to recover full names in the training data from learned word embeddings. Dziedzic et al. (2023) note that *stealing* sentence encoders by distilling through API queries works well and is difficult for API providers to prevent. Song and Raghunathan (2020) considered the problem of recovering text sequences from embeddings, but only attempted to recover bags of words from the embeddings of a shallow encoder model. Li et al. (2023) investigate the privacy leakage of embeddings by training a decoder with a text embedding as the first embedding fed to the decoder. Compared to these works, we consider the significantly more involved problem of developing a method to recover the full ordered text sequence from more realistic state-of-the-art text retrieval models.

**Gradient leakage.** There are parallels between the use of vector databases to store embeddings and the practice of federated learning, where users share gradients with one another in order to jointly train a model. Our work on analyzing the privacy leakage of text embeddings is analogous to research on *gradient leakage*, which has shown that certain input data can be reverse-engineered from the model's gradients during training (Melis et al., 2018; Zhu et al., 2019; Zhao et al., 2020; Geiping et al., 2020). Zhu et al. (2019) even shows that they can recover text inputs of a masked language model by backpropagating to the input layer to match the gradient. However, such techniques do not apply

to textual inversion: the gradient of the model is relatively high-resolution; we consider the more difficult problem of recovering the full input text given only a single dense embedding vector.

**Text autoencoders.** Past research has explored natural language processing learning models that map vectors to sentences (Bowman et al., 2016). These include some retrieval models that are trained with a shallow decoder to reconstruct the text or bag-of-words from the encoder-outputted embedding (Xiao et al., 2022; Shen et al., 2023; Wang et al., 2023). Unlike these, we invert embeddings from a frozen, pre-trained encoder.

## 9 Conclusion

We propose Vec2Text, a multi-step method that iteratively corrects and re-embeds text based on a fixed point in latent space. Our approach can recover 92% of 32-token text inputs from their embeddings exactly, demonstrating that text embeddings reveal much of the original text. The model also demonstrates the ability to extract critical clinical information from clinical notes, highlighting its implications for data privacy in sensitive domains like medicine.

Our findings indicate a sort of equivalence between embeddings and raw data, in that both leak similar amounts of sensitive information. This equivalence puts a heavy burden on anonymization requirements for dense embeddings: embeddings should be treated as highly sensitive private data and protected, technically and perhaps legally, in the same way as one would protect raw text.

## 10 Limitations

**Adaptive attacks and defenses.** We consider the setting where an adversary applies noise to newly generated embeddings, but the reconstruction modules were trained from un-noised embeddings. Future work might consider reconstruction attacks or defenses that are adaptive to the type of attack or defense being used.

**Search thoroughness.** Our search is limited; in this work we do not test beyond searching for 50 rounds or with a sequence beam width higher than 8. However, Vec2Text gets monotonically better with more searching. Future work could find even more exact matches by searching for more rounds with a higher beam width, or by implementing more sophisticated search algorithms on top of our corrective module.

**Scalability to long text.** Our method is shown to recover most sequences exactly up to 32 tokens and some information up to 128 tokens, but we have not investigated the limits of inversion beyond embeddings of this length. Popular embedding models support embedding text content on the order of thousands of tokens, and embedding longer texts is common practice (Thakur et al., 2021). Future work might explore the potential and difficulties of inverting embeddings of these longer texts.

**Access to embedding model.** Our threat model assumes that an adversary has black-box access to the model used to generate the embeddings in the compromised database. In the real world, this is realistic because practitioners so often rely on the same few large models. However, Vec2Text requires making a query to the black-box embedding model for each step of refinement. Future work might explore training an imitation embedding model which could be queried at inference time to save queries to the true embedder.

## 11 Acknowledgements

Thanks to Simran Arora and Piotr Teterwak for helpful conversations about the research. JXM is supported by a NSF GRFP. VK is supported by NSF 2145577 and NIH MIRA R35GM151243. VS is partially supported by NSF 1916717. AMR is supported by NSF 2242302, NSF CAREER 2037519, IARPA HIATUS, and a Sloan Fellowship.

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

# A Appendix

## A.1 Additional analysis

**How does word frequency affect model correctness?** fig. 6 shows the number of correct predictions (orange) and incorrect predictions (blue) for ground-truth words, plotted across word frequency in the training data. Our model generally predicts words better that are more frequent in the training data, although it is still able to predict correctly a

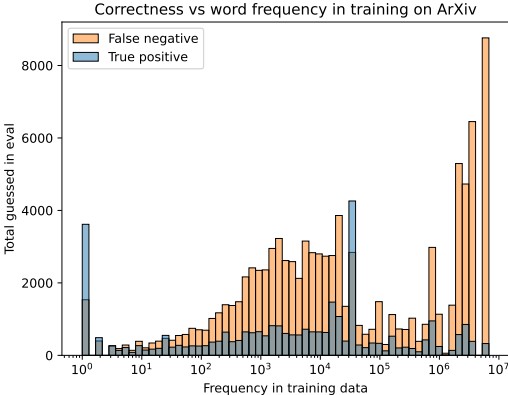

Figure 6: Correctness on evaluation samples from ArXiv data.

number of words that were not seen during training[3]. Peaks between $10^4$ and $10^5$ come from the characters (, $-$, and ), which appear frequently in the training data, but are still often guessed incorrectly in the reconstructions.

## A.2 Full defense results

Results on each dataset from BEIR under varying levels of Gaussian noise are shown in Appendix A.2. The model is GTR-base. Note that the inputs are limited to $32 tokens$, far shorter than the average length for some corpuses, which is why baseline ($\lambda = 0$) NDCG@10 numbers are lower than typically reported. We included the full results (visualized in Figure 2) as Appendix A.2.

---

[3]We hypothesize this is because all test *tokens* were present in the training data, and the model is able to reconstruct unseen words from seen tokens.

| $\lambda$ | arguana | bioasq | climate-fever | dbpedia-entity | fiqa | msmarco | nfcorpus | nq | quora | robust04 | scidocs | scifact | signal1m | trec-covid | trec-news | webis-touche2020 |
|---|---|---|---|---|---|---|---|---|---|---|---|---|---|---|---|---|
| 0 | 0.328 | 0.115 | 0.136 | 0.306 | 0.208 | 0.647 | 0.239 | 0.306 | 0.879 | 0.205 | 0.095 | 0.247 | 0.261 | 0.376 | 0.245 | 0.233 |
| 0.001 | 0.329 | 0.115 | 0.135 | 0.307 | 0.208 | 0.647 | 0.239 | 0.306 | 0.879 | 0.204 | 0.096 | 0.246 | 0.261 | 0.381 | 0.246 | 0.233 |
| 0.01 | 0.324 | 0.113 | 0.132 | 0.301 | 0.205 | 0.633 | 0.234 | 0.298 | 0.875 | 0.192 | 0.092 | 0.235 | 0.259 | 0.378 | 0.234 | 0.225 |
| 0.1 | 0.005 | 0.000 | 0.000 | 0.000 | 0.000 | 0.000 | 0.017 | 0.000 | 0.003 | 0.000 | 0.002 | 0.006 | 0.001 | 0.005 | 0.001 | 0.000 |
| 1.0 | 0.001 | 0.000 | 0.000 | 0.000 | 0.000 | 0.000 | 0.008 | 0.000 | 0.000 | 0.000 | 0.000 | 0.001 | 0.000 | 0.000 | 0.000 | 0.000 |

Table 6: BEIR performance (NDCG@10) for GTR-base at varying levels of noise (32 tokens).

| $\lambda$ | NDCG@10 | BLEU |
|---|---|---|
| 0.000 | 0.302 | 80.372 |
| 0.001 | 0.302 | 72.347 |
| 0.010 | 0.296 | 10.334 |
| 0.100 | 0.002 | 0.148 |
| 1.000 | 0.001 | 0.080 |

Table 7: Retrieval performance and reconstruction performance across varying noise levels $\lambda$.