# OpenReview forum: "Text Embeddings Reveal (Almost) As Much As Text"
_EMNLP/2023/Conference — EMNLP 2023 Main_

### Official Review · Reviewer_WRMs · 2023-07-27

**Soundness:** 4

**Excitement:**

5: Transformative: This paper is likely to change its subfield or computational linguistics broadly. It should be considered for a best paper award. This paper changes the current understanding of some phenomenon, shows a widely held practice to be erroneous in someway, enables a promising direction of research for a (broad or narrow) topic, or creates an exciting new technique.

**Paper Topic And Main Contributions:**

This paper aims to investigate the information stored in text embeddings generated by text encoders and specifically focuses on recovering the original full texts from the text embeddings. The authors formulate the task as controlled text generation and adopt a multi-step approach that gradually refines the hypothesis text closer to the given target embedding.

The proposed method demonstrates that most short texts (e.g., 32-token) can be reconstructed almost exactly. Such findings show that the private information in the training data could be leaked from the output text embeddings of language models.

**Questions For The Authors:**

* It is mentioned in Line 290 that "More rounds is monotonically helpful"; is this guaranteed (theoretically)? If so, can we perfectly recover every sequence eventually given infinite rounds of refinement?
* I'd be curious to know what kinds of raw texts are more challenging than others to recover (i.e., failure cases or cases that require many rounds of refinement).

**Reasons To Accept:**

* Novelty: The paper appears to be very novel to me, from the task it considers (i.e., extracting information as raw texts from text embeddings), to the proposed method (i.e., multi-step refinement in controlled text generation), to the insights that follow (i.e., text embeddings may leak private information). I believe the paper is among the most novel papers I've seen at NLP venues.
* Clarity: The paper is clear in motivation, method description, and experiment settings.
* Interesting and effective approach: The proposed method Vec2Text is reasonable, interesting, and effective in extracting the original texts from the embeddings. It significantly outperforms naive and standard decoding approaches (e.g., beam search and nucleus sampling). The authors also propose a simple method for defending against inversion attacks by injecting some random noise into the embeddings.
* Comprehensive evaluations: The evaluation is very comprehensive, covering many different tasks and domains. The case studies are good analyses for understanding the model's property.
* Important implications: The insights brought by the paper could have important implications for how language models should be configured so as to preserve the private/sensitive information in the raw text data.


**Reasons To Reject:**

* The method efficiency (as inversion attack) could be a bit concerning, considering that it adopts a multi-step refinement approach. However, based on my understanding, the major contribution of the paper is not proposing an attack approach, but rather to show that there exist such approaches that may cause privacy concerns regarding leaking out text embeddings. Therefore, I won't consider this as a weakness of the paper.

Overall, I don't have any major concerns about the paper that prevent it from being accepted.

**Reproducibility:**

3: Could reproduce the results with some difficulty. The settings of parameters are underspecified or subjectively determined; the training/evaluation data are not widely available.

**Reviewer Confidence:**

4: Quite sure. I tried to check the important points carefully. It's unlikely, though conceivable, that I missed something that should affect my ratings.

---

> ### Author Rebuttal · Authors · 2023-08-29
>
> Thank you for your comments. We sincerely appreciate your critique and address some of your concerns below.
>
> > It is mentioned in Line 290 that "More rounds is monotonically helpful"; is this guaranteed (theoretically)?
>
> We can only guarantee that the corrected text won’t get further from the initial text by only “accepting” a change if it improves in cosine similarity to the ground-truth embedding. However, it is possible (and common) to “get stuck” and fail to find a correction that improves the current text.
>
> We feel this is an interesting question and there is room here to develop random methods (MCMC-style) that avoid this phenomenon of getting stuck and perhaps improve recovery of the ground-truth text after many rounds of refinement.
>
> > I'd be curious to know what kinds of raw texts are more challenging than others to recover (i.e., failure cases or cases that require many rounds of refinement).
>
> To answer this question, we analyze 100 reconstructions from Natural Questions after 1 and 50 rounds of refinement. At a single round, we see reconstructions are generally on topic but do not match the syntax of the true sequence and miss or hallucinate many entities. Entities gradually “snap into place” after more rounds of refinement, and we see 94/100 sequences perfectly recovered.
>
> Out of all 6 sequences that are not recovered after 50 rounds of refinement, we see that the predicted sequence is very close in every case, and contains all of the correct entities, but has swapped two entities or clauses. For example, given the ground truth:
>
> ```decision to have the series focus on Patrick and Robin's romance, Ed Martin of the "Jack Myers Report" stated, "...it’```
>
> Vec2Text finds the reconstruction:
>
> ```decision to have the series focus on Patrick and Robin's romance, Ed Myers of the "Jack Martin Report" stated, "...it’```
>
> Notice that the entities are correct, but the last names “Myers” and “Martin” are swapped. We hypothesize this common failure mode may be due to the bag-of-words information conveyed in common sentence embeddings.

---

### Official Review · Reviewer_Qu6b · 2023-08-04

**Typos Grammar Style And Presentation Improvements:** The size of Table 4 should be adjusted.
**Soundness:** 3

**Excitement:**

3: Ambivalent: It has merits (e.g., it reports state-of-the-art results, the idea is nice), but there are key weaknesses (e.g., it describes incremental work), and it can significantly benefit from another round of revision. However, I won't object to accepting it if my co-reviewers champion it.

**Paper Topic And Main Contributions:**

This paper shows that embeddings itself are able to be restored as its origin text.
As a frame of embedding inversion, the author presents their method Vec2Text, which uses encoder-decoder architecture as well as a new input format. The input format consists of projected original embedding, hypothesis embedding, the difference of them, word embeddings.
The performances on several datasets reach around 90%, as the authors said, the observation could make privacy issue.

**Questions For The Authors:**

- Why do you think the method--adding Gaussian noise are effective?
- During the iterative process, I think the hyperparameters should be adjusted.

**Reasons To Accept:**

- As the authors mentioned, since a lot of systems use vector embeddings and sometimes we can access the embedding itself, the problem that the author tackled is interesting.
- Proposed Vec2Text is an iterative simple method.
- After showing the issue, the author suggested a simple method to prevent it.

**Reasons To Reject:**

- Although the paper report interesting findings, I cannot clarify the main contribution.
- As far as I understand, the contribution seems to be more into their proposed method to restore the text.
- If so, more analysis and ablations are required for the evaluation
  - For small length of tokens, the performance is much better compared to the baseline, but I cannot find the explanation.
  - Discussion on Table 4 and Table 5 is missing
- Furthermore, there is no clear reason for not comparing with other embedding inversion methods and the methods mentioned in related work section; because of this, I cannot convince how much Vec2Text is effective.
  - Without reasonable baseline, people could consider the performance as a result of the further training process.
- On the other hand, the Gaussian noise method weakens the contribution of vec2text because the method is not robust on the noise, which means not generalized well.

In conclusion, I personally feel the findings interesting but the presentation of current version is weak.

**Reproducibility:**

3: Could reproduce the results with some difficulty. The settings of parameters are underspecified or subjectively determined; the training/evaluation data are not widely available.

**Reviewer Confidence:**

3: Pretty sure, but there's a chance I missed something. Although I have a good feel for this area in general, I did not carefully check the paper's details, e.g., the math, experimental design, or novelty.

---

> ### Author Rebuttal · Authors · 2023-08-29
>
> Thank you for your comments.
>
> > Although the paper report interesting findings, I cannot clarify the main contribution.
>
> Our main contribution is Vec2Text, a method for iteratively recovering sequences from their embeddings. We show that with a known embedder, this can cause privacy leakage.
>
> > For small length of tokens, the performance is much better compared to the baseline, but I cannot find the explanation.
>
> We looked into this question further and we believe it was due to an optimization. We recently shifted to initializing model parameters from scratch instead of initializing from the zero-step inversion model. After this change we are able to train our model for many more tokens without overfitting.
>
> This resulted in Vec2Text achieving the following performance reconstructing text from OpenAI embeddings of MSMARCO of up to 128 tokens:
>
> |              method               |  token f1  |  bleu  |  exact | cos |
> | ---- | ---- | ---- | ---- | ---- |
> | Base                   |  54 |   17.0  |    0.6  |  0.95 |
> | Vec2Text               |        |       |         |       |
> | --> [1 step]            |  68 | 29.9   |    1.4  |  0.97 |
> | --> [20 steps]          |  78 | 43.1  |    3.2  |  0.99 |
> | --> [50 steps]          |  78 | 44.4   |    3.4  |  0.99 |
> | --> [50 steps + sbeam]   |   84 |  55.0  |    8.0  |  0.99 |
>
> With longer training, new long-sequence Vec2Text model significantly outperforms the one from the original draft of our paper (55.0 vs. the originally reported BLEU of 33.9), showing a similar relative improvement.
>
> Due to the fixed-size embedding, we do expect lower absolute recovery performance for longer sequences. In the future, we could achieve better results using Vec2Text on the same embeddings by scaling model parameters or using more compute at search time.
>
> > Discussion on Table 4 and Table 5 is missing.
>
> Thanks for noticing the lack of references here – we do discuss these results although we do not directly reference the tables from the text. We will fix in a future version.
>
> > there is no clear reason for not comparing with other embedding inversion methods and the methods mentioned in related work section
>
> We were not able to find many alternative baseline approaches. We do include a bag-of-words baseline in Table 1, which was proposed in Song (2022).
>
> The only other baseline from related work (Adolphs et al (2022) and Li et al (2023)) trains a decoder model with the projected sentence embedding as the first token; we expect this to perform weaker than our proposed architecture, an encoder-decoder with sequence projection. To compare, we trained this model on NQ using a GPT-2 backbone and will add this result to a future version. Results show this architecture performs far worse than our baseline:
>
> |              method               |  token f1  |  bleu  |  exact | cos
> | ---- | ---- | ---- | ---- | ---- |
> | New baseline (GPT-2)              |        47  |   1.0  |      0 | 0.76
> | Existing baseline (T5 + MLP2Seq)  |        67  |  31.9  |      0 | 0.91
> | Vec2Text                          |        99  |  97.3  |     92 | 0.99
>
> > Why do you think the method--adding Gaussian noise are effective?
>
> Our hypothesis is that adding Gaussian noise seems to move embedding vectors out of training distribution. As the noise factor λ increases, noisy embeddings look less like the training data, causing the inversion methods to fail.
>
> > During the iterative process, I think the hyperparameters should be adjusted.
>
> Vec2Text has two hyperparameters: number of iterations and sequence-level beam width. We provide results at 0, 1, 20, and 50 iterations, and with sequence-level beam widths of 1 and 8.

---

### Official Review · Reviewer_3tQ6 · 2023-08-12

**Soundness:** 3

**Excitement:**

4: Strong: This paper deepens the understanding of some phenomenon or lowers the barriers to an existing research direction.

**Paper Topic And Main Contributions:**

The manuscript "Text Embeddings Reveal (Almost) As Much As Text" presents a framework that is able to reconstruct a 32-token input text from dense embeddings.

Main contributions:
- Addressing privacy concerns for vector databases;
- Thorough investigation of the embedding inversion task leveraging embedding geometry;
- Controlled Generation approach able to recover up to 92% of input text.

**Questions For The Authors:**

Major:
- Consider embeddings from bert-like models (mainly used in the clinical space for privacy concerns) as input for the MIMIC dataset.
- Report a better investigation of the types of best-reconstructed sentences and the information they carry for each dataset used.

**Reasons To Accept:**

The work is methodologically sound and the experiments are thorough. It includes: embedding inversion experiments, ablation studies, Gaussian noise addition as a way to fix the problem, and the impact of word frequency investigation. Given the recent availability of generative language models that can be directly queried in inference, the topic of privacy concerns related the generation of sensitive text is central and need further investigation. This applies especially in cases where underrepresented populations are at risk of harm and in the clinical space.

**Reasons To Reject:**

Despite its relevance, the work mainly highlights the performance of the approach and to a less extent the actual harm that can come from different tasks. The authors leveraged several benchmark datasets but do not report a thorough characterization of the extent of harm that the embedding inversion can have for each dataset. In particular, authors use the MIMIC dataset to prove that names and sensitive information can be retrieved. Nevertheless, they use 32-token text that is not a comparable length when dealing with clinical text where each note can be thousand of tokens long. Privacy concerns in the healthcare space are currently addressed by using encoder-only models and deidentified clinical notes, hence the experiment performed by the authors on MIMIC does not seem particularly relevant.

**Reproducibility:**

3: Could reproduce the results with some difficulty. The settings of parameters are underspecified or subjectively determined; the training/evaluation data are not widely available.

**Reviewer Confidence:**

3: Pretty sure, but there's a chance I missed something. Although I have a good feel for this area in general, I did not carefully check the paper's details, e.g., the math, experimental design, or novelty.

**Typos Grammar Style And Presentation Improvements:**

Some typos in the text and Related Works section should go after Introduction for clarity and readability.

---

> ### Author Rebuttal · Authors · 2023-08-29
>
> Thank you for your comments.
>
> > Privacy concerns in the healthcare space are currently addressed by using encoder-only models  ... Consider embeddings from bert-like models.
>
> We trained an additional base and corrector model given the ClinicalBERT encoder (Huang et al (2019)). Here we report its performance on recovering information from MIMIC notes:
>
> method | first | last | full | bleu | tf1 | exact | cos
> |---|---|---|---|---|---|---|---|
> Base | 40.0 | 28.1 | 12.5 | 3.9 | 31.7 | 0.0 | 77.1
> Vec2Text | 91.7 | 84.2 | 75.4 | 43.8 | 75.1 | 0.2 | 96.8
>
> These experiments show that embeddings from BERT-based models can be successfully inverted. We also plan to add full results on NQ for this ClinicalBERT model to the appendix.
>
> > they use 32-token text that is not a comparable length when dealing with clinical text where each note can be thousand of tokens long
>
> Our work is the first to show that text sequences of *any* length can be exactly recovered given just their embeddings, indicating that embedding inversion is a threat to privacy. We run other experiments on text up to 128 tokens. We leave it to future work to investigate how inversion performs with much longer embedded sequences.
>
> Even small, highly incomplete fractions of clinical notes reveal very sensitive information, such as general medical area (eg, oncology), details of diagnosis (eg, type of tumor), some aspects of the treatment regimen, etc.
>
> > Privacy concerns in the healthcare space are currently addressed by using encoder-only models and deidentified clinical notes
>
> We note that GTR, the embedder we use for MIMIC, *is* an encoder-only model as it uses the encoder-part of T5. We also note the above results from ClinicalBERT, another encoder is analyzed in this rebuttal.
>
> We also note that while deidentification is one defence, there are multiple examples of re-identification attacks against medical datasets that were de-identified to the HIPAA standards, for example:
> - Sweeney, Latanya. “Matching Known Patients to Health Records in Washington State Data”. https://papers.ssrn.com/sol3/papers.cfm?abstract_id=2289850
> - Yoo, Ji Su, Alexandra Thaler, Latanya Sweeney, and Jinyan Zang. "Risks to Patient Privacy: A Re-identification of Patients in Maine and Vermont Statewide Hospital Data." Technology Science (October 2018)
>
> Deidentification provides very brittle protection at best.  If embeddings can be reversed to deidentified notes, these notes become very vulnerable to reidentification.
>
> > Report a better investigation of the types of best-reconstructed sentences and the information they carry for each dataset used.
>
> We performed manual error analysis on 100 examples from the Natural Questions dataset, of which Vec2Text (with 50 iterations) only fails on 6 of them. It appears that reordering two similar entities was the most common failure mode for our model. Text with few or easily distinguished entities appears easier to reconstruct.

---

### Official Review · Reviewer_a8TL · 2023-08-13

**Soundness:** 5

**Excitement:**

5: Transformative: This paper is likely to change its subfield or computational linguistics broadly. It should be considered for a best paper award. This paper changes the current understanding of some phenomenon, shows a widely held practice to be erroneous in someway, enables a promising direction of research for a (broad or narrow) topic, or creates an exciting new technique.

**Paper Topic And Main Contributions:**

This paper explores the possibility of reconstructing full-text content from dense text embeddings. The authors cast this problem as a controlled generation task, aiming to generate text that, when re-embedded, closely matches a fixed point in the latent space. While a naive model conditioned on the embedding performs poorly, the authors propose a multi-step method, Vec2Text, which iteratively corrects and re-embeds text, and achieves an astonishing accuracy in recovering text inputs exactly. Experiments on various datasets across different domains (i.e., NQ, MSMARCO, MIMIC-III, and 15 datasets from BEIR) support the major claim of this paper: text embeddings reveal almost as much as text. Moreover, to defend against such inversion attacks, the authors also study the possibility of adding moderate noises to the embeddings, which, according to their experiments, significantly decreases the reconstruction accuracy without hurting the retrieval performance too much.

**Questions For The Authors:**

- Could you run each experiment multiple times, report the mean and standard deviation, and conduct statistical significance tests to compare Vec2Text with the strongest baseline in each column?

- Could you report the performance of Base and other Vec2Text variants on BEIR?

- Instead of adding noises to the embeddings to defend against inversion attacks, I am thinking about a simpler solution: just using some less accurate embedding models (e.g., those ranked 10th or 15th on MTEB). Their retrieval performance should be on par with SOTA embedding models with a certain level of noise, but is it possible that they enjoy an even lower reconstruction accuracy? I understand this is a complex question and needs a thorough empirical study, so I do not expect a perfect answer from the authors, but I suggest they add this to their discussion.

**Reasons To Accept:**

+ The observations in this study have important implications for several popular issues, such as privacy in large language models and vector databases.

+ The empirical study is comprehensive. Various retrieval-related benchmark datasets are used, covering both in-domain and cross-domain reconstruction settings. Two state-of-the-art text embedding models (based on T5 and GPT, respectively) are considered. A case study is also presented on MIMIC-III in the medical domain. The Analysis section is inspiring as well.

+ Based on the empirical observations, the authors also propose a simple yet effective way to defend against inversion attacks to text embeddings.

**Reasons To Reject:**

- It is unclear how the reconstruction performance will be affected by randomness. The authors should run each experiment multiple times with the mean and standard deviation reported. Also, significance tests are missing. It is unclear whether the improvement of Vec2Text[50 steps+sbeam] over baselines and other Vec2Text variants is statistically significant or not. In fact, some gaps of tf1 and cos in Table 1 are quite subtle, therefore p-values should be reported.

- It is unclear whether Vec2Text[50 steps+sbeam] also outperforms baselines and other Vec2Text variants on cross-domain datasets, such as BEIR.

**Reproducibility:**

4: Could mostly reproduce the results, but there may be some variation because of sample variance or minor variations in their interpretation of the protocol or method.

**Reviewer Confidence:**

3: Pretty sure, but there's a chance I missed something. Although I have a good feel for this area in general, I did not carefully check the paper's details, e.g., the math, experimental design, or novelty.

---

> ### Author Rebuttal · Authors · 2023-08-29
>
> Thank you for your comments.
>
> > It is unclear how the reconstruction performance will be affected by randomness. Could you run each experiment multiple times, report the mean and standard deviation...
>
> The numbers reported in the paper are always taken from the mean of 1000 test datapoints. We plan to add standard errors in the final version. Here are results on Natural Questions with standard errors:
>
> | method | bleu | token f1 | exact | cos
> | ----- | ----- | ----- | ----- | ----- |
> | Base (0 steps) | 29.1 ± 1.2 | 65.1 ± 0.8 | 1.5 ± 0.6 | 90.0 ± 0.3
> | Vec2Text | | | |
> | --> [1 step] | 47.7 ± 2.9 | 78.0 ± 0.6 | 9.7 ± 1.3 | 95.1 ± 0.2
> | --> [20 steps] | 81.3 ± 1.1 | 94.4 ± 0.4 | 51.8 ± 2.2 | 98.8 ± 0.1
> | --> [50 steps] | 82.9 ± 1.0 | 95.2 ± 0.3 | 52.4 ± 2.2 | 99.0 ± 0.1
> | --> [50 steps + sbeam] | **97.0** ± 1.3 | **99.1** ± 0.5 | **94.0** ± 2.3 | **99.9** ± 0.0
>
> > ...and conduct statistical significance tests?
>
> To confirm the difference in BLEU scores between Vec2Text and the baseline, we perform a paired t-test on with a confidence level of 0.95 and a null hypothesis that the BLEU scores of our method is equivalent to the baseline. We collect 384 BLEU scores from running the baseline and Vec2Text on reconstructing 32-token GTR-embedded inputs from NQ. We observe a t-statistic of -27.9 and a p-value of 9.82e-103, much smaller than the significance level of 0.05. Thus, we reject the null hypothesis and conclude that the performance of our method exceeds the baseline with a confidence level of 95%.
>
> > It is unclear whether Vec2Text[50 steps+sbeam] also outperforms baselines and other Vec2Text variants on cross-domain datasets, such as BEIR.
>
> Here we report the results of Vec2Text (using the new and improved model) compared to the baseline across the datasets of BEIR, measuring performance by BLEU score. We see that Vec2Text provides on average 3.4x performance improvement across these out-of-domain test datasets:
>
> | dataset          |   Baseline |   Vec2Text |
> |-----------------|-----------|-----------|
> | arguana          |        6.8 ± 1.1 |    **20.6 ± 0.9** |
> | bioasq           |      6.7 ± 0.9   |       **22.8 ± 3.0** |
> | climate-fever    |       12.8 ± 1.7 |       **44.9 ± 3.3** |
> | cqadupstack      |        7.1 ± 1.6 |      **24.2 ± 0.2** |
> | dbpedia-entity   |       15.4 ± 2.2 |       **45.3 ± 2.6** |
> | fever            |       12.6 ± 1.1 |       **45.1 ± 0.5** |
> | fiqa             |        6.6 ± 0.6 |      **21.6 ± 1.8** |
> | hotpotqa         |       15.4 ± 2.6 |       **38.9 ± 1.1** |
> | msmarco          | 15.5 ± 0.8 | **59.6 ± 4.0** |
> | nfcorpus         |        6.2 ± 0.33 | **26.1 ± 1.9** |
> | nq               |       11.0 ± 0.9 | **32.7 ± 1.0** |
> | quora            |       36.2 ± 1.8 | **95.6 ± 2.9** |
> | robust04         |     4.6 ± 0.2  |  **15.5 ± 0.6** |
> | scidocs          |      5.5 ± 0.3   | **17.2 ± 1.1** |
> | scifact          |        5.0 ± 0.4 |  **18.1 ± 1.4** |
> | signal1m         |       13.2 ± 1.5 | **72.3 ± 2.2** |
> | trec-covid       |        5.6 ± 0.6 |  **19.2 ± 0.4** |
> | trec-news        |        4.9 ± 0.2 |  **14.5 ± 0.5** |
> | webis-touche2020 |        6.6 ± 0.5 | **19.6 ± 0.5** |
>
>
> > Instead of adding noises to the embeddings to defend against inversion attacks, I am thinking about a simpler solution: just using some less accurate embedding models.
>
> This is a good idea for a possible defense: swapping the embedder (regardless of accuracy) should severely inhibit the performance of our system. (Note that this is out of our threat model, as we assume that our system trains on embeddings from a particular embedder, we do not expect to generalize to other embedders.)
>
> We the effect of changing the embedder on performance by replacing embeddings from GTR with those from ClinicalBERT using our model trained on GTR embeddings, and observe the following inversion performance:
>
>
> method | bleu | token f1 | exact | cos
> | ---- | ---- | ---- | ---- | ---- |
> Base (0 steps) | 1.0 ± 0.5 | 6.4 ± 0.3 | 0.0 ± 0.0 | 48.6 ± 0.8
> Vec2Text | | | |
> --> [1 step] | 1.8 ± 0.0 | 12.3 ± 0.0 | 0.0 ± 0.0 | 52.0 ± 0.4
> --> [20 steps] | 1.7 ± 0.1 | 11.6 ± 0.0 | 0.0 ± 0.0 | 65.5 ± 0.7
> --> [50 steps] | 1.7 ± 0.1 | 11.6 ± 0.5 | 0.0 ± 0.0 | 65.5 ± 0.7
> --> [50 steps + sbeam] | 1.6 ± 0.0 | 10.6 ± 0.7 | 0.0 ± 0.0 | 65.6 ± 0.6
> [Gaussian noise + GTR + Vec2Text] | 1.9 ± 0.0 | 10.6 ± 0.3 | 0.0 ± 0.0 | 1.8 ± 0.0
>
> For reference, we include an additional row showing the performance of our GTR model given the proper embedder under multivariate Gaussian noise with a factor of 1.0. Even these essentially random generations achieve a token F1 score of 10.6 by simply outputting common words.
>
> When we substitute the embedder with GTR with ClinicalBERT, performance severely drops: exact match performance drops from 94% to 0. Vec2Text barely outperforms random language model generations in token-match metrics, although iteration does slightly help in locating sequences with higher similarity to the ground-truth.

---

### Meta-Review · Area_Chair_9Mcq · 2023-09-19

**Recommendation:** 5

**Metareview:**

This article aims to study the information stored in text embeddings generated by text encoders, with a particular focus on restoring the original full text from text embeddings. The author describes this task as controlled text generation and adopts a multi-step method to gradually refine the hypothetical text to make it closer to the given target embedding. The proposed method indicates that most short texts (such as 32 tokens) can be almost accurately reconstructed. These findings indicate that private information in the training data may have been leaked from the output text embedding of the language model.In summary, these paper solve privacy issues in vector databases， conduct in-depth research on embedding inversion tasks using embedded geometry and he controlled generation method can recover up to 92% of input text. Meanwhile, this paper's experiments are very comprehensive such as various retrieval-related benchmark datasets are used and covering domain including in-domain and closed-domain reconstruction settings.

---

### Decision · Program_Chairs · 2023-10-07

**Decision:**

Accept-Main

**Comment:**

This article aims to study the information stored in text embeddings generated by text encoders, with a particular focus on restoring the original full text from text embeddings. The author describes this task as controlled text generation and adopts a multi-step method to gradually refine the hypothetical text to make it closer to the given target embedding. The proposed method indicates that most short texts (such as 32 tokens) can be almost accurately reconstructed. These findings indicate that private information in the training data may have been leaked from the output text embedding of the language model.In summary, these paper solve privacy issues in vector databases， conduct in-depth research on embedding inversion tasks using embedded geometry and he controlled generation method can recover up to 92% of input text. Meanwhile, this paper's experiments are very comprehensive such as various retrieval-related benchmark datasets are used and covering domain including in-domain and closed-domain reconstruction settings.